# Validation of an Eye-Tracking Algorithm Based on Smartphone Videos: A Pilot Study

**DOI:** 10.3390/diagnostics15121446

**Published:** 2025-06-06

**Authors:** Wanzi Su, Damon Hoad, Leandro Pecchia, Davide Piaggio

**Affiliations:** 1School of Engineering, University of Warwick, Library Road, Coventry CV4 7AL, UK; wanzi.su@warwick.ac.uk; 2Warwick Medical School, University of Warwick, Gibbet Hill Road, Coventry CV4 7AL, UK; damon.hoad@swft.nhs.uk; 3Department of Engineering, University Campus Bio-Medico of Rome, Via Alvaro del Portillo 21, 00128 Rome, Italy; l.pecchia@warwick.ac.uk

**Keywords:** eye-tracking, eye movement, neurodegenerative condition, biosignal processing

## Abstract

**Introduction:** This study aimed to develop and validate an efficient eye-tracking algorithm suitable for the analysis of images captured in the visible-light spectrum using a smartphone camera. **Methods:** The investigation primarily focused on comparing two algorithms, which were named CHT_TM and CHT_ACM, abbreviated from the core functions: Circular Hough Transform (CHT), Active Contour Models (ACMs), and Template Matching (TM). **Results:** CHT_TM significantly improved the running speed of the CHT_ACM algorithm, with not much difference in the resource consumption, and improved the accuracy on the x axis. CHT_TM achieved a reduction by 79% of the execution time. CHT_TM performed with an average mean percentage error of 0.34% and 0.95% in the x and y direction across the 19 manually validated videos, compared to 0.81% and 0.85% for CHT_ACM. Different conditions, like manually opening the eyelids with a finger versus without a finger, were also compared across four different tasks. **Conclusions:** This study shows that applying TM improves the original eye-tracking algorithm with CHT_ACM. The new algorithm has the potential to help the tracking of eye movement, which can facilitate the early screening and diagnosis of neurodegenerative diseases.

## 1. Introduction

Eye-tracking is an increasingly popular methodology across various applications. What captivates numerous researchers is its time-efficiency, cost-effectiveness, reliability, objectiveness, quantifiability, easiness to conduct, non-invasiveness, and accessibility. By leveraging techniques such as shape-based, appearance-based, feature-based, and hybrid methods, eye-tracking technologies capture and analyse the movements of key ocular components, including the pupil, cornea, sclera, iris, and retina [1,2,3]. These movements rely on the physiological mechanism controlled by the central nervous system, involving the brainstem, cerebellum, basal ganglia, and cerebral cortex [4]. Consequently, eye-tracking measurements provide a valuable window into brain function and cognitive processes.

Eye movements are categorised into four basic types: saccades, smooth pursuit movements, vergence movements, and vestibulo-ocular movements [5]. Even during fixation, the eye was not completely still due to fixational eye movements, which are composed of three main kinds of smaller movements: microsaccades, drift, and tremor [6]. Microsaccades and drifts have an average size of approximately 6′ arc, with drifts exhibiting an average velocity of about 1′ arc per second [7]. These eye movements can influence each other and, sometimes, occur simultaneously. For example, in Miki and Hirata’s study [8], microsaccades were observed in parallel with the ongoing vestibulo-ocular reflex. Among the aforementioned eye features, saccades and fixations are commonly analysed in eye-tracking studies [9], although pursuit and pupil dilation are also frequently investigated [10,11,12].

A systematic literature review has summarised the eye-tracking methods for enhancing productivity and reading abilities [13], categorising studies into four areas: general eye metrics and tracking, comprehension measurement, attention measurement, and typography and typesetting. Notably, the eye metrics such as fixation duration, saccade velocity, and blink rates have been linked to reading behaviours including speed reading and mind-wandering. Although findings across studies have limited generalizability, the integration of machine learning with eye-tracking technologies has shown promising potential in understanding reading behaviour, attention, and cognitive workload.

### 1.1. Eye-Tracking in Healthcare

Eye-tracking has demonstrated significant potential in healthcare applications, particularly in telemedicine [6], assistive technologies (e.g., wheelchair control [14]), and the assessment of neurological and vestibular disorders such as acute-onset vertigo [15]. It has been used to evaluate memory in conditions such as Amyotrophic Lateral Sclerosis (ALS) and late stages of Alzheimer’s Disease (AD), without relying on intact verbal or motor functions, which may not prevail or be reliable in patients of late stages. By minimising the need for complex instructions or decision-making, eye-tracking provides an accessible means of evaluation [9].

Eye-tracking features have been identified as potential biomarkers for multiple neurodegenerative disorders. For instance, impaired convergence is observed in Parkinson’s disease (PD) [16], while atypical saccades characterise preclinical and early-stage Huntington’s disease [17,18]. Neurological abnormalities could also be detected in long COVID patients. García Cena et al. proved that long COVID patients were affected by altered latency for centrally directed saccades, impaired memory-guided saccades, and increased latencies for antisaccades [19].

### 1.2. Challenges and Limitations

Despite these promising applications, there are still concerns and challenges. The experimental constraints and demands of certain tasks can lead to changes in microsaccade behaviour, potentially producing responses that do not occur in natural conditions. For example, Mergenthaler and Engbert observed more microsaccades in the fixation task than the naturalistic scene viewing task, suggesting those responses might be task-specific [20]. However, in real-world settings, human behaviour will involve a range of different tasks (with both fixation and free viewing present), resulting in a combination of different types of eye movements. This highlights the difficulty of replicating experimental conditions outside controlled laboratory settings with the use of special eye-tracking equipment.

Therefore, barriers to the widespread adoption of these examinations in daily life or at home require patients to visit a clinic or laboratory for eye-tracking assessments. Moreover, the expensive price and high-calibration requirements, which need specific training, present further obstacles [6,21,22,23]. While some methods are accurate and effective, they often require skilled professionals to operate, making the process laborious and time-consuming [24].

To address these limitations, researchers have explored the use of consumer-grade electronic devices like smartphones, video recorders, laptops, and tablets. However, these approaches introduce new challenges, particularly regarding head stabilisation, which significantly affects tracking accuracy. Solutions such as external head restraints and head-mounted equipment have been proposed to mitigate these issues [6,21,25]. Another solution is to use some indicators (e.g., a sticker) to represent the head movement and then subtract it from the eye movement [22].

### 1.3. State of the Art

Eye-tracking is sometimes combined with functional magnetic resonance imaging (fMRI) to investigate neurological deficits [26,27]. Compared to fMRI, one of the advantages of automated eye feature evaluation is its ability to simultaneously collect multiple measures during the same session and obtain different information for specific analysis and diagnosis [9]. High-frame-rate cameras further enhance detection capabilities, revealing some covert saccades that are difficult to see with the naked eye.

In order to collect and synthesise the available evidence, in a chapter of ‘Eye Movement Research’, Hutton [28] introduced the range of eye-tracking technologies currently available. There are two common tests to measure eye movement: video recording (video-oculography or VOG) and electromyography (electro-oculography or EOG). The less common approaches are through limbus reflection, Dual Purkinje Image (DPI), and scleral search coils, but they will not be discussed in detail here.

EOG uses a pair of electrode patches and measures the activity of two cranial nerves, which, respectively, connect the brain with the inner ear and the eye muscles. VOG uses a camera to record the eye movements and calibrate the accuracy. According to Corinna Underwood [29], there is a general consensus that the accuracy of directly measuring eye movements is higher than indirectly measuring movements via eye muscle motion. Hutton further highlights that EOG’s primary drawback is its susceptibility to spatial inaccuracies due to drift artefacts over time, caused by impedance changes in the electrodes [28]. While most studies favour VOG over EOG, there is insufficient evidence to conclusively recommend one method over the other. VOG, though more precise, is relatively expensive and requires the eyes to remain open. Meanwhile, using EOG requires careful consideration in practice as certain medications (e.g., sedatives) or medical electrical equipment (e.g., cardiac pacemakers) would affect or interfere with the electromyography function [29].

VOG systems require head stability due to vestibulo-ocular movements that cause involuntary eye motion (to maintain the gaze fixation on a location, the head motion would cause the eye movement in the opposite direction [6]). Solutions to this issue fall into two categories: stationary VOG, where the camera remains fixed and external supports such as chin rests stabilise the head, and mobile VOG, where the camera moves with the head (e.g., in wearable eye-tracking glasses). Mobile VOG is generally better suited for real-world tasks, whereas stationary VOG is typically used in research settings that require specific stimuli. However, the need for mobile VOG to be lightweight and wearable imposes limitations on its specifications. As a result, its data quality, sampling rates, and spatial resolution are generally lower than those of high-specification stationary VOG systems [28].

Infrared-based eye-tracking, known as infrared oculography (IOG), is a common VOG method that tracks corneal reflection and the pupil centre [9,30,31]. Under infrared illumination, the pupil appears as a comparatively dark circle compared to the rest of the eye (e.g., the iris and sclera) [28]. However, this approach assumes that the pupils dilate and constrict symmetrically around their centre, which is not always accurate under varying luminance conditions according to Wildenmann et al. [32]. Consequently, researchers continue to explore alternative methods that do not rely on infrared illumination.

Table 1 provides a comparative analysis of different eye-tracking technologies, including fMRI, VOG, and EOG, highlighting their respective advantages and limitations [6,9,28,29].

### 1.4. Deep Learning Methods and Hardware Limitations

Deep learning, particularly convolutional neural network (CNN) methods, has become a popular approach in modern eye-tracking research due to its capacity to learn direct mappings from raw facial and ocular images to gaze coordinates. These methods have been greatly accelerated by the availability of large-scale gaze datasets (GazeCapture [33] and MPIIGaze [34]) and the outstanding performance of CNNs in computer vision tasks.

Several CNN architectures have been explored in the literature. Early approaches include different CNN structures such as LeNet-based [35], AlexNet-based [36], VGGNet-based [34], and ResNet18-based [37] and ResNet50-based [38] models. Among notable CNN-based models, the iTracker model by Krafka et al. [33] used facial and eye images to predict gaze with high accuracy and was trained using the GazeCapture dataset. Valliappan et al. [39] demonstrated the feasibility of smartphone-based gaze tracking. Their proposed multi-layer feed-forward CNN achieved a gaze estimation accuracy of 0.6–1°, which is comparable to commercial solutions such as Tobii Pro Glasses.

Despite their effectiveness, these models typically demand a high computational burden, which limits their applicability for real-time deployment on mobile devices. They often require CPUs and GPUs with high cost and power consumption. Due to hardware limitations such as constrained computational power, limited battery life, and network bandwidth, deep learning models are not well suited for low-resource settings and edge computing, particularly on standard smartphone devices.

To address these limitations, several lightweight CNN architectures have been proposed, such as SqueezeNet [40] and ShuffleNet [41]. Notably, MobileNet [42] and MobileNet-V2 [43] employ depthwise convolutions to significantly reduce parameter count and computational cost while maintaining reasonable accuracy. When combined with edge computing platforms such as the Raspberry Pi, Google Coral USB Accelerator, and NVIDIA Jetson TX2, this direction appears promising.

However, these lightweight models are typically only suitable for gaze estimation on specific smartphone models with relatively high hardware specifications, even with support from mobile-optimised frameworks such as TensorFlow Lite (TFLite) and PyTorch Mobile. While this may become increasingly viable with ongoing advancements in smartphone technology and reductions in device cost, such solutions are not yet feasible in low-resource settings. In these contexts, a traditional processing algorithm embedded within the smartphone may offer a more accessible and practical solution. Ultimately, the choice reflects a trade-off between the required accuracy for the research question and the limitations of the target hardware.

### 1.5. Smartphone-Based Eye-Tracking

Modern smartphones present an excellent opportunity for accessible and cost-effective eye-tracking. Equipped with high-resolution cameras and increasing on-device processing power, they are well-suited for edge computing, i.e., bringing the computation closer to the devices where data is gathered rather than cloud computing. This allows smartphones to function as both data acquisition and processing units, enabling real-time analysis, reducing latency, and preserving user privacy, which are particularly valuable in medical diagnostics. Their internet connectivity further supports remote deployment and monitoring when needed. Previous research has demonstrated the feasibility of using smartphones as an alternative to commercial-grade equipment for pupillometry [23]. Their affordability, ease of use, and resilience to power supply make them an attractive solution for large-scale, low-resource applications.

One challenge with smartphone-based tracking is its small screen size, which can restrict visual stimulus presentation. Many researchers have addressed this by incorporating external displays. For instance, Azami et al. [44] used an Apple iPad display and an Apple iPhone camera to record eye movements in ataxia and PD participants. After comparing different machine learning methods, they decided to use principal components analysis (PCA) and linear support vector machine (SVM), achieving 78% accuracy in distinguishing individuals with and without oculomotor dysmetria. Similarly, before switching to tablets, Lai et al. [21] initially designed their experiment based on a laptop display and an iPhone 6 camera. The iTracker-face algorithm can estimate mean saccade latency with a precision of less than 10 ms error. However, they did not report gaze accuracy. One exception is the study by Valliappan et al. [39], using a smartphone (Pixel 2 XL) as both a display and camera on healthy subjects.

While there is a wide variety of equipment and algorithms, few studies integrate eye-tracking with edge computing. Gunawardena et al. [45] found only one study (by Otoom et al. [46]) that combined edge computing with eye-tracking. As for the accuracy, Molina-Cantero et al. [47] reviewed 40 studies on visible-light eye-tracking using a low-cost camera, reporting an average visual angle accuracy of 2.69 degrees. Among these, only a subset achieved errors below 2.3 degrees, including the studies by Yang et al. [48], Hammal et al. [49], Liu et al. [50,51], Valenti et al. [52], Jankó and Hajder [53], Wojke et al. [54], Cheng et al. [55], and Jariwala et al. [56].

### 1.6. Contribution

Given the gap between the considerable potential of smartphones and the lack of edge computing solutions in eye-tracking, the proposed algorithm aims to introduce a lightweight alternative to conventional resource-demanding deep learning neural networks. As highlighted in the literature review, this cross-modal, smartphone-based approach is computationally efficient. Its low requirements make it especially suitable for smartphones and its compatibility with edge computing supports deployment in low-resource settings, where conventional systems are impractical or unavailable.

Although most studies would unintentionally mix the concept of eye-tracking and gaze estimation, it is important to distinguish between these two methods. Eye-tracking involves analysing visual information from images or frames in VOG, focusing on identifying facial landmarks, the eye region, and the pupil centre. In contrast, gaze estimation further requires stimulus calibration, head orientation detection, and gaze angle calculation. This article aims to develop a stationary VOG application that prioritises eye-tracking over gaze tracking, utilising smartphone cameras as an alternative to infrared-based systems.

This study evaluates the performance of two eye-tracking algorithms—Circular Hough Transform plus Active Contour Models (CHT_ACM) and Circular Hough Transform Template Matching (CHT_TM)—in tracking the iris, particularly in neurodegenerative conditions. The findings will inform the second phase of the study, aimed at developing a novel gaze estimation software that is based on images acquired in the visible-light spectrum and will be validated against a commercial infrared eye tracker.

Modern eye trackers in gaming (e.g., Tobii Eye Tracker 5 [57] or Gazepoint 3 Eye Tracker [58]) achieve an optimistic performance in gaze tracking with unconstrained head movements. However, in line with the United Nations’ Sustainable Development Goal 3 (Good Health and Wellbeing) and the principle of frugal innovation, this study aims to develop a cost-effective, smartphone-based eye-tracking solution that eliminates the need for additional accessories, thereby improving accessibility in low-resource settings. This study further contributes to the ongoing development of smartphone-based eye-tracking, paving the way for wider adoption in both clinical and non-clinical settings.

## 2. Methods

### 2.1. Experiment Definition

In order to build a dataset to develop and try the proposed algorithm on, an ethical approval application was submitted at the Biomedical and Scientific Research Ethics Committee of the University of Warwick and was granted (Ethical approval number 08/22-23).

This protocol was developed under the supervision of a medical doctor, who was an expert in neurorehabilitation, and was tested on three subjects with different iris colours, to investigate the robustness of the proposed algorithm. The three subjects had brown, green, and gray eyes, respectively. The experiment took place in the same laboratory setting and the environmental illumination was measured by a Dr.meter LX1010B digital lux meter (DrMeter, Union City, CA, USA).

During the experiment, the subject was also recorded in two different conditions, i.e., one with the eyelids naturally maintained open, and one where they were asked to keep them open by using their fingers. This was done to understand the influence of blinks and eyelids’ partial closure on the algorithm performance. The experiment stimuli were designed to have an orange circle as the target and a dotted line to show the trajectory (shown in Figure 1). There were four tasks in total: (a) vertical task; (b) horizontal task; (c) fixation task; and (d) circular task. The target would follow the target along the trajectory only once, i.e., one complete round of the circle and back and forth for the linear tasks. The duration of each movement was either 1 or 5 s in order to mimic both slow and fast type of movements (e.g., smooth pursuit vs. nystagmus). The participant was asked to use fingers to open the eyelids, after one session without doing so.

The equipment preparation involved merely a smartphone (iPhone 12, Apple Inc., Cupertino, CA, USA, 1920 × 1080 pixels, 30 frames/s), a desktop display (Samsung Electronics Co., Ltd., Suwon, South Korea), and a tripod (PEYOU Ltd., Wan Chai, Hong Kong). The participant was asked to sit in a private room in the author’s department with a desktop screen right in front of them and a smartphone slightly shifted aside to avoid blocking the screen. The participant was asked to perform these tests without any other tool including the use of chin rest, which is normally used for aiding them to keep their head still. The participants were asked to follow certain instructions on the screen in front, while the different aforementioned tasks were being shown. In the experiment, the smartphone tracker placed on a tripod was used to track and record the eye movement of the participants. The detailed experimental set-up with the relevant quantifications is displayed in Figure 2.

### 2.2. Algorithm Development

Starting from the acquired videos, which featured the whole face, and after excluding the invalid videos (i.e., those with interfering movements) and manually cropping them to the region of interest (ROI), i.e., the whole eye, the first step in algorithm development entailed the selection of the best pipeline for image pre-processing and iris detection. At this stage, the working platform was Python 3.9.12, and the OpenCV (4.6.0) library was used for most functions. In terms of image pre-processing, the sampling rate for extracting the frames from the video was every 10 frames. Morphological operations [59] (i.e., erosion and dilation) and binarisation were applied. Erosion was used to reduce the influence of the eyelashes on the iris, and dilation was applied to remove the light spots caused by the reflection of light. Consequently, the CHT [60] was applied (minDist: 40; param1: 180; param2: 10; minRadius: 15; maxRadius: 50) in two different cases, one on data pre-processed as above plus ACM, and one on data pre-processed as above plus TM.

CHT is an image analysis technique designed to detect circular shapes within an image, even when the circles are partially occluded or incomplete. It operates by identifying radial symmetry through the accumulation of “votes” in a three-dimensional parameter space defined by the circle’s centre coordinates and radius. Each edge pixel in the input image contributes votes in a circular pattern for potential circle centres. When multiple edge pixels support a circle of the same radius and centre, the votes accumulate at those coordinates. The peaks or local maxima in the voting space indicate the presence and location of the centroid. In real-world scenarios, such as eye-tracking, the pupil often appears elliptical due to the angle between the eye and the camera. Consequently, the voting space may produce a cluster of local maxima rather than a single point, as illustrated in Figure 3. This effect can be mitigated by adjusting the voting threshold and defining a minimum distance between detected peaks.

CHT on its own, despite several adjustments to its parameters to reach optimal performance, was not powerful enough, resulting in too many eligible circle candidates to be the actual iris as output. In order to select the correct circle, binarisation [62] was then applied. This allowed for the removal of other spurious round circles. As mentioned above, ACM was also applied as a pre-processing step for one of the algorithms. ACM [63] works by aligning a deformable model with an image through energy minimisation. This deformable spine, also called a snake, responds to both constraint and image forces. These forces act collaboratively, pulling the snake toward object contours, while internal forces counteract undesired deformations. Specifically, snakes are tailored to address scenarios where an approximate understanding of the boundary’s shape is available.

TM [64], conversely, represents an advanced machine vision technique designed to discern portions of an image that correspond to a predefined template. The process involves systematically placing the template over the image at all conceivable positions, calculating a numerical measure of similarity between the template and the currently overlapped image segment each time. Subsequently, the positions yielding the highest similarity measures are identified as potential occurrences of the template. The computation of the similarity measure between the aligned template image and the overlapped segment of the input image relies on the cross-correlation technique, entailing a straightforward summation of pairwise multiplications of corresponding pixel values from the two images.

Our CHT_TM relies on CHT_ACM only for the first frame. In detail, ACM is used in the first frame to fit an approximate boundary initiated with a circular snake inside of the ROI. With this approximate boundary, the image could be further cropped in order to reduce the noise and enhance the performance. CHT provides the potential circle candidates and binarisation selects the best iris circle. After obtaining an intact iris template through CHT_ACM, TM is applied in the following frames to find the position with the highest similarity. The position is translated into the corresponding iris centre and is validated against the manual measurements.

### 2.3. Algorithm Validation and Error Avoidance

To quantify the speedup, the running speed was calculated by measuring the running time of the same algorithm on the data of 10, 40, 100, and 400 frames on the same video. Moreover, in order to perform an initial validation of the algorithm, manual measurements were taken every 10 frames (same as algorithm development) on the 19 acquired videos from 9 different stimuli. Manually annotating every frame of the videos is not efficient. These stimuli represented various conditions that needed to be analysed, including finger/no finger and quick/slow movement.

Manually annotated iris centres were used as ground truth and comparison for the output of the algorithm. Several measurements of performance including mean absolute error (MAE), mean percentage error (MPE), root mean square error (RMSR), and Pearson Correlation Coefficient (PCC) were calculated by the Euclidean distance between the actual location and ground truth in both x and y directions. For a given set of coordinates on the x axis as an example, x_actual_ is the ground truth, x_measure_ is the output of the algorithm, and n is the number of the frames. Here are the mathematical equations of the metrics:(1)$$MAE=\frac{1}{n}\sum_{i=1}^{n}\left|x_{actual,i}-x_{measure,i}\right|$$(2)$$MPE=\frac{1}{n}\sum_{i=1}^{n}\frac{\left|x_{actual,i}-x_{measure,i}\right|}{x_{actual,i}+EPSILON}*100\%$$(3)$$RMSE=\sqrt{\frac{1}{n}\sum_{i=1}^{n}{{(x}_{actual,i}-x_{measure,i})}^{2}}$$(4)$$Pearson\;r=\frac{\sum_{i=1}^{n}\left(x_{actual,i}-{\bar{x}}_{actual})(x_{measure,i}-{\bar{x}}_{measure}\right)}{\sqrt{{\sum_{i=1}^{n}\left(x_{actual,i}-{\bar{x}}_{actual}\right)}^{2}}\sqrt{{\sum_{i=1}^{n}\left(x_{measure,i}-{\bar{x}}_{measure}\right)}^{2}}}$$

Once Pearson Correlation Coefficient r is calculated, the t-statistic is then computed:(5)$$t=r\cdot \sqrt{\frac{n-2}{1-r^{2}}}$$

Assuming t~t_n−2_, the *p*-value is calculated using the survival function (two-tailed):(6)$$pval=2\cdot sf\left(\left|t\right|\right)\;\;$$

This validation process compared the difference between several conditions including ones with/without fingers (to keep the eyelid open), 1 s/5 s movement, with/without TM, and all types of eye movements (fixation, vertical, horizontal, circle).

The frames in which the subject was blinking were deemed invalid. Moreover, due to the tendency of CHT_TM to match the lower half of the non-intact eye (shown in Figure 4a) resulting in a larger y axis error, a masking technique (shown in Figure 4b) that only calculates the pixels within the central circle was applied. The larger error on y axis (still quite a low error) also represented the technical issue while developing CHT_TM, i.e., that CHT_TM always tended to match the lower half of the non-intact eye (shown in Figure 4a).

## 3. Results

A total of 19 videos were acquired from 3 subjects, resulting in a total of 633 analysed frames. The illumination was natural daylight with an average intensity of 1680 ± 282.49 lux. The first subject performed 9 videos with 288 images in total, and the other two subjects performed 5 videos with 173 images in total.

Figure 5 describes the steps of the algorithm. The valid frames of the videos were cropped into the eye region and converted into grayscale. After that, the first frame was processed by CHT_ACM. An intact iris template was generated, and the iris centre position of the first frame was calculated. With this template, the following frames were processed by CHT_TM, which output the iris centre positions of the frames. The results could be calculated accordingly, and the performance was validated with manual measurements (see Figure 5).

### 3.1. CHT_ACM Versus CHT_TM

Figure 6 compares the two different algorithms: CHT_ACM and CHT_TM. The CHT_ACM method begins with Active Contour Models to define a region of interest, followed by histogram-based binarisation of the image. It then applies the Circular Hough Transform to detect eligible circles and evaluates them by summing white pixel values in their respective binarised regions. A circle is identified as the iris if its white pixel values are below a 60% threshold. In contrast, the CHT_TM method uses a predefined iris template for Template Matching and identifies locations with a similarity metric above a threshold. For each candidate, it calculates the pixel value within a circular region and selects the darkest (lowest value) location as the final iris centre.

The TM function would significantly increase the running speed of the CHT_ACM (as shown in Figure 7). The average running time per frame (seconds) is 1.298 (CHT_ACM) and 0.271 (CHT_TM). CHT_TM saved 79% of the execution time, which means it is around 5 times faster, with a good trade-off between execution time and resource use. Most of the resource consumption is on reading the image so CPU and memory usage is not much different. Therefore, Template Matching greatly improved the running speed of CHT_ACM.

The MAE of CHT_TM was 1.43 pixels and 1.75 pixels in the x and y axis (shown in Table 2). Meanwhile, the MAE of CHT_ACM was 1.77/2.08 for x/y axis, less accurate than CHT_TM. CHT_TM performed with an average MPE of 1.06% and 1.21% in the x and y direction across the 19 videos, compared to 1.29% and 1.42% for CHT_ACM. Moreover, Pearson’s Correlation Coefficients for CHT_TM were extremely high (r_x_ of 0.993, p value_x_ less than 0.001; r_y_ of 0.990, p value_y_ less than 0.001). The conclusion is that Template Matching not only improved the running speed but also the accuracy of CHT_ACM.

### 3.2. With-Fingers Versus Without-Fingers Condition

In order to investigate the influence of using fingers to open the eyelid, the results of two conditions were analysed, i.e., with fingers or without fingers. In Table 3, it is obvious that the with-finger condition has lower error and higher Pearson Correlation Coefficient value, except one row, namely “x_CHT_TM”. This suggests that using fingers can improve the performance of CHT_ACM and CHT_TM on the y axis, but it reduces the accuracy of CHT_TM on the x axis. PCC_p values of Table 3 are all below 0.00001. Therefore, these values are not reported in Table 3.

### 3.3. Comparison Among Tasks

Table 4 presents the performance of the two algorithms CHT_ACM and CHT_TM on the four tasks, i.e., vertical task, horizontal task, circular task, and fixation task. This is used to analyse the characteristics of each algorithm on different tasks, which helps identify the specific improvements of CHT_TM compared to CHT_ACM. Moreover, it provides evidence of what is sacrificed by applying TM. It is worth noticing that CHT_ACM performs better on the x axis in the fixation task but is worse in the horizontal task and vertical task regardless of axis. On the other hand, CHT_TM is more accurate on the y axis but has larger error on the x axis in the circular task and fixation task. PCC_p values of Table 4 are all below 0.00001. Therefore, these values are not reported in Table 4.

The best and the worst performing applications for both algorithms are summarised in Table 5. The fixation task is the most accurate on the y axis regardless of CHT_ACM or CHT_TM and the PCC is the lowest. The horizontal task can be well tracked on the x axis by CHT_TM but is the worst for CHT_ACM no matter the axis. The vertical task is best measured on the x axis by CHT_ACM. The circular task has the largest error in general, which means it is the hardest to track, especially by CHT_TM. However, the PCC of the circular task is the highest so the direction is similar between the real output and the manual validation result. PCC_p values of Table 5 are all below 0.00001. Therefore, these values are not reported in Table 5.

### 3.4. Comparison Among Iris Colours

To understand the robustness of the proposed algorithm, the error metrics from subjects with different iris colours are presented in Table 6. The performance table indicates that there is a slight decrease in the accuracy of the algorithm for subjects with light iris colour but the final error is still pixel-level. Specifically, subject 1 with brown iris colour had the lowest error, especially on x axis, and subject 2 with green iris colour had the highest error. Subject 3 with gray iris colour showed a higher error on x axis than y axis, which was different from the other two subjects. The accuracy of CHT_TM was better than CHT_ACM across all three subjects.

## 4. Discussion

This study was aimed at developing and validating an effective eye-tracking algorithm to be used on visible-light images captured by a smartphone camera, in order to unlock more affordable and user-friendly technologies for eye-tracking. Two algorithms, named CHT_TM and CHT_ACM, were compared in terms of performance and computational efficiency. The selection of these algorithms was based on their potential to enhance accuracy and speed without increasing resource consumption.

Comprehensively, CHT_TM demonstrated improved runtime and superior performance in vertical eye movement tracking (y axis), although CHT_ACM outperformed it in horizontal tracking (x axis) in two out of four tasks. Whether it was CHT_ACM or CHT_TM, larger errors were consistently observed along the y axis, as seen in Table 2. This can be attributed to the anatomical reality that the upper and lower regions of the iris are more likely to be covered by the eyelids, especially during upward or downward gaze, or when participants are fatigued and the eyes are half-closed. This introduces inaccuracies in iris centre detection.

From the results comparing with/without-fingers conditions, using the fingers to open the eyelid seems to improve accuracy in most cases. This supports the hypothesis that using fingers can help the algorithm diminish the error of non-intact iris as the eyelid is no longer covering the iris. While CHT_TM performed better along the x axis even without finger assistance, the benefit of finger usage was more significant on the y axis, where eyelid interference is typically greater on the upper edge or the lower edge. Despite this, the use of fingers was reported as uncomfortable for participants and is therefore not advisable in future studies. Alternative non-invasive strategies or postprocessing solutions are recommended.

Task-wise, the circular task produced the largest errors, suggesting that the primary challenge lay in the instability of the task rather than the algorithm itself. CHT_ACM remains less accurate than CHT_TM. As the mean absolute errors of both CHT_ACM and CHT_TM were on the pixel level, these errors were overall very small, also when taking into account the limits of the system for manual measurements with a precision of 0.5 pixels. Fixation tasks, being the most stable, resulted in the lowest tracking errors for both algorithms, which means there was no drastic head movement during the experiment. CHT_TM was better at tracking the fixation task on the y axis. Interestingly, CHT_TM improved x axis tracking in the horizontal task, likely due to its robustness in recognising elliptical iris shapes during lateral gaze. In contrast, CHT_ACM retained an advantage on the x axis for the fixation task. CHT_TM performed more reliably across tasks, especially in preventing tracking loss.

Comparing subjects with different iris colour, the algorithm showed the best performance in subjects with dark iris colour. Nonetheless, both algorithms produced pixel-level errors across all subjects, with CHT_TM consistently outperforming CHT_ACM. This indicates that the iris colour would influence the accuracy of both algorithms simultaneously, especially on the x axis.

No notable differences were found between the fast and the slow experiment.

Despite minor differences in error rates, it is noteworthy that both algorithms achieved high accuracy, with an average error of just 1.7 pixels (1.2%) across 19 videos. These results underscore the feasibility of both methods for reliable iris centre detection. However, the most significant difference lay in computational performance, with CHT_TM offering faster processing times.

Comparing the proposed method with existing smartphone-based approaches proved challenging due to a lack of validated benchmarks and methodological transparency in the literature. Many studies fail to disclose algorithmic details and rely instead on vague references to platforms like OpenCV or ARKit, thereby hindering reproducibility. In contrast, this study prioritises transparency and reproducibility by making both the data and algorithms publicly available.

Some may question the absence of machine learning or deep learning in this work, especially given their strong performance in image analysis tasks. However, the lack of a suitable public dataset, particularly one containing data from neurodegenerative patients, prevented the use of AI-based models. The mismatch between public eye-tracking datasets and cognitive experiment settings makes it difficult to adopt or adapt open-access datasets, considering the final goal of distinguishing patients and healthy subjects.

Public datasets are typically based on static stimuli and short-duration recordings, and are primarily designed to optimise gaze point accuracy. In contrast, cognitive experiments involve dynamic, goal-directed, task-specific stimuli, capture temporal dependencies in sequential data, and focus on variables linked to cognitive processes such as attention, memory, and response inhibition. Moreover, datasets collected in clinical or cognitive contexts are smaller due to stricter requirements and limited participant availability. These datasets are also subject to privacy and ethical regulations, which restrict data sharing and limit opportunities for large-scale model training. Additionally, variations in experimental conditions such as viewing distance, screen size, inter-subject variability (patients vs. controls), and ambient lighting hinder experimental reproducibility and limit the transferability of pre-trained models. In fact, data collection from patients would still be inevitable, but replicating the same experiment settings as the used public dataset would remain challenging as explained above.

Additionally, using AI trained on different hardware and image conditions (e.g., infrared cameras) would compromise compatibility with the smartphone-based set-up employed here. For instance, the structure of this algorithm is inspired by and is similar to the one proposed by Zhang et al. [65]. However, they used a CNN to condense the video and their data was collected from a portable infrared video goggle instead of the smartphone intended in this paper. Not only is the video grayscale but also the distance from the camera to the eye is different, making it impossible to use their data in this experiment or to develop the same algorithm based on varied data.

Beyond data limitations, deploying AI models on smartphones presents practical challenges. Deep learning methods typically require powerful processors or graphics processing units (GPUs) designed for computer systems, which is not the best fit for smartphones. In this case, it is necessary to upload the video data to the cloud server and use cloud computing. This reliance introduces new issues such as internet connectivity (not always available in rural areas or within low-resource settings), delayed response times, and potential data privacy risks (identifiable data like face videos). In contrast, a self-contained, built-in algorithm avoids these complications and better serves low-resource environments.

Nonetheless, AI remains a promising avenue for future work. Studies have shown a similar or even better performance with a CNN using the front-facing camera of Pixel 2 XL phone [39] or the RealSense digital camera [66] than a commercial eye tracker. Once a sufficiently large and diverse dataset is collected, including both healthy individuals and patients, AI models could be trained to refine or replace parts of the current algorithm. Such models could automate pre-processing or minimise tracking errors through learned feature extraction.

The future plan for this study is to develop a refined experimental protocol in collaboration with medical professionals, followed by validation against a commercial infrared eye tracker. Then, experiments can be carried out at hospital level on patients affected by neurodegenerative conditions. The ultimate goal is to design a smartphone-compatible eye-tracking toolkit and AI-based system for the early screening of neurodegenerative diseases.

### Limitations

One limitation of the current protocol is the visible trace of the target during the task (see Figure 1), which allows participants to predict the target’s trajectory. As a result, their eye movements may precede rather than follow the target. Additionally, the absence of a headrest introduces variability due to head movement, which can compromise signal quality. To address this, future studies will explore the feasibility of using a sticker placed in a fixed location as a reference point to track and compensate for head motion, allowing reconstruction of more accurate eye movement data.

Although the study aligns with the principle of frugal innovation and avoids using additional apparatus, a tripod was used as a temporary substitute for a user’s hand or arm. One thing worth noticing is the differences in participant height, which can affect the camera angle towards the eye and may contribute to varying errors among subjects.

Another limitation is the small sample size, as this pilot study was primarily intended to demonstrate feasibility. Manual validation was used to assess the algorithm’s performance. This method, while effective for small datasets, lacks the efficiency and scalability of automated validation methods. This limits the generalizability of the results and the potential for large-scale application.

At this stage, comparisons were made between algorithm outputs and manual annotations of actual eye movement centres, rather than estimated gaze points. Each video was relatively short and did not include significant head movement, so the ROI of the eye was manually cropped and fixed at a constant image coordinate. Consequently, all movement was referenced to the same top-left corner of the cropped ROI (coordinate [0, 0]).

## 5. Conclusions

This paper shows that applying TM improves the original eye-tracking algorithm with CHT_ACM. The improved algorithm shows potential for supporting eye movement tracking in early screening and diagnosis of neurodegenerative diseases. Its integration into a non-invasive, cost-effective telemedicine toolbox could improve diagnostic reliability and accessibility as well as electronic transferability, in a cost-effective way. Such innovations sit perfectly within the current healthcare landscape, which has limited funding and is challenged by health emergencies (e.g., COVID-19 pandemic). Our research contributes to the development of such telemedicine tools by offering a reliable and accessible method for eye-tracking, which can be integrated into telemedicine platforms for remote screening and diagnosis. As highlighted above, the study has several limitations including reliance on a traditional image processing approach rather than AI, visibility of the target trace, and the lack of head movement correction. Future work will aim to refine the experimental protocol with guidance from medical experts, validate the algorithm against commercial benchmarks, and expand testing to patients with neurodegenerative symptoms. The integration of AI techniques will also be explored as a potential enhancement to the system.

## Figures and Tables

**Figure 1 diagnostics-15-01446-f001:**
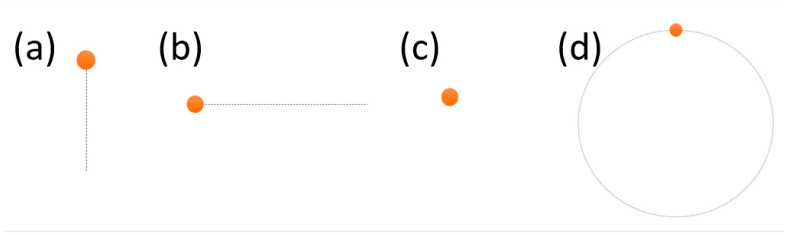
The initial set of the stimuli in the experiment protocol. (**a**) Vertical task; (**b**) horizontal task; (**c**) fixation task; and (**d**) circular task.

**Figure 2 diagnostics-15-01446-f002:**
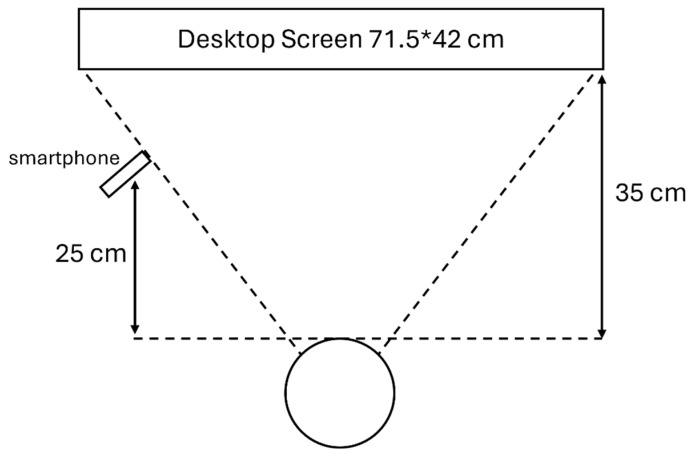
The experimental set-up.

**Figure 3 diagnostics-15-01446-f003:**
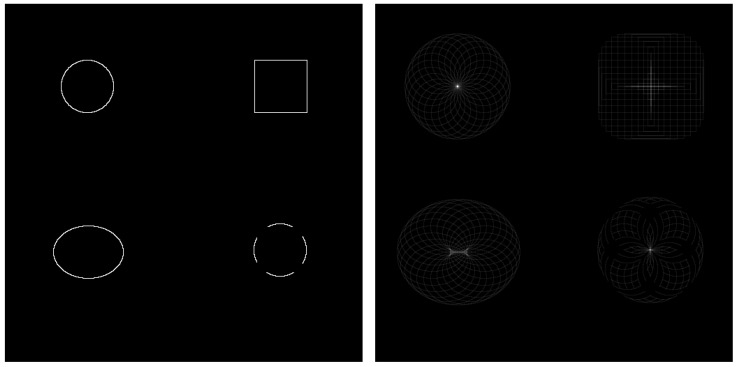
The voting mechanism from the edge pixel in a circular pattern [61].

**Figure 4 diagnostics-15-01446-f004:**
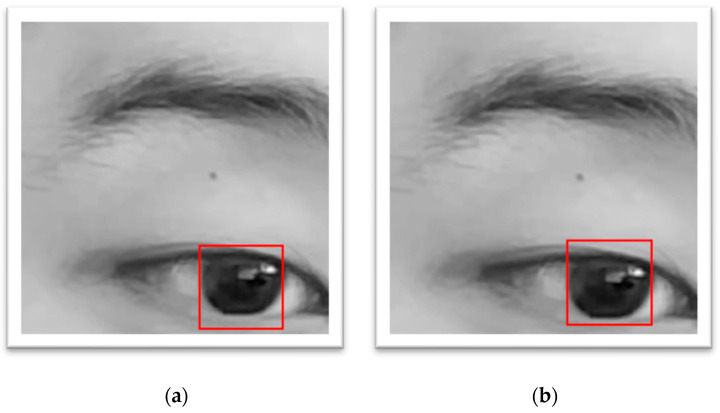
An example of matching the lower half of non-intact eye. (**a**) Before using masking. (**b**) After using masking.

**Figure 5 diagnostics-15-01446-f005:**
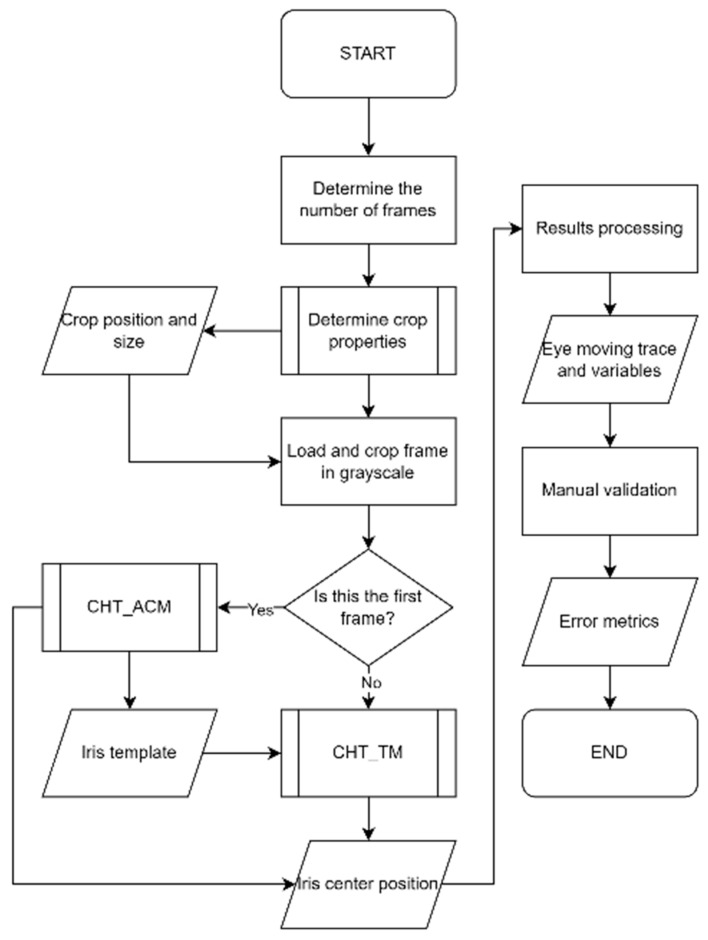
The flowchart describes the steps of the algorithm.

**Figure 6 diagnostics-15-01446-f006:**
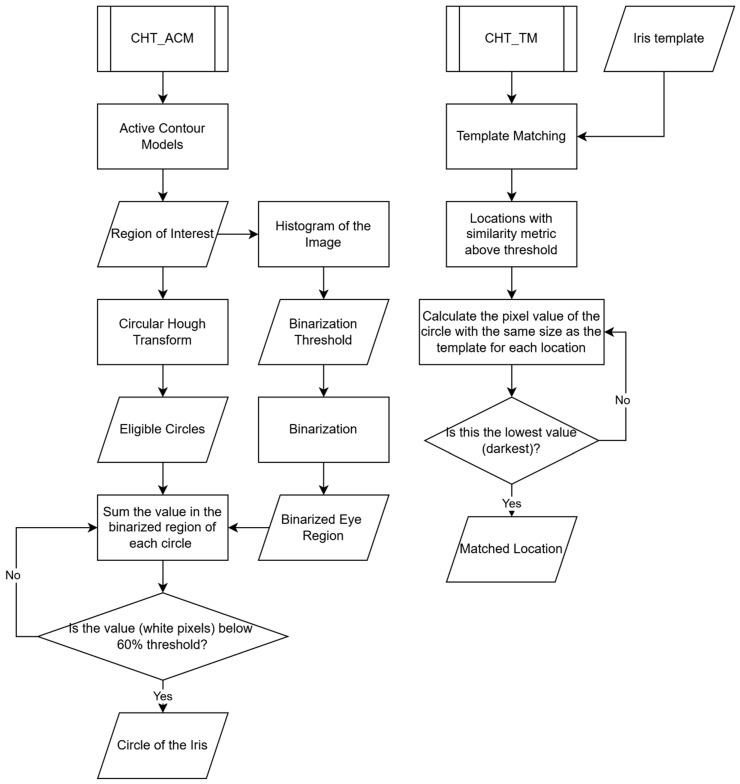
The flowchart comparing the two different algorithms in this study.

**Figure 7 diagnostics-15-01446-f007:**
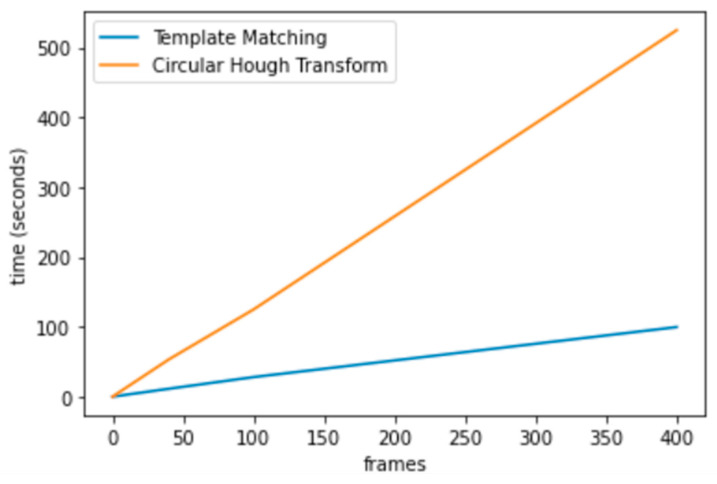
The running speed of CHT_ACM and CHT_TM.

**Table 1 diagnostics-15-01446-t001:** A comparison between the advantages and disadvantages of different state-of-the-art technologies.

Technologies			Advantages	Disadvantages
**fMRI and other technology**			For specific purposes	Without multiple eye measures
**Eye-Tracking**	**EOG**		Cheap, well established, readily available, high temporal resolution, no need for eyes to be open	Low accuracy and safety
	**VOG**	**Stationary**	High specifications	Need stimuli (likely in research scenarios)
		**Mobile**	No need for head stability, real-world tasks, lightweight	Low data quality, low sampling rates and poor spatial resolution
		**Infrared Cameras (pupil centre)**	Not influenced by the environment light levels	Pupil change error (pupils do not dilate or constrict symmetrically around their centre)

**Table 2 diagnostics-15-01446-t002:** Mean average error, mean percentage error, root mean square error, and Pearson Correlation Coefficient calculated in both x and y directions of all 19 videos. * denotes better performance (lower errors).

	CHT_TM	CHT_ACM
MAE for x	1.432 *	1.774
MAE for y	1.752 *	2.079
MPE for x	1.059 *	1.290
MPE for y	1.213 *	1.423
RMSE for x	2.362 *	2.514
RMSE for y	2.554 *	3.069
Pearsons r and pval for x	0.993, <0.0001	0.993, <0.0001
Pearsons r and pval for y	0.990, <0.0001	0.986, <0.0001

**Table 3 diagnostics-15-01446-t003:** The comparison of with-fingers and without-fingers conditions. * denotes better performance (lower errors) between with-fingers and without-fingers conditions.

No Finger	MAE	MPE	RMSE	PCC_r
x_CHT_TM	0.467 *	0.304 *	0.748 *	0.999
y_CHT_TM	1.697	1.241	2.396	0.984
x_CHT_ACM	1.36	0.904	1.997	0.995
y_CHT_ACM	1.337	0.969	1.876	0.990
**Finger**	**MAE**	**MPE**	**RMSE**	**PCC_r**
x_CHT_TM	0.541	0.403	0.849	0.999
y_CHT_TM	0.739 *	0.476 *	0.995 *	0.997
x_CHT_ACM	0.901 *	0.653 *	1.252 *	0.998
y_CHT_ACM	1.009 *	0.653 *	1.760 *	0.993

**Table 4 diagnostics-15-01446-t004:** The comparison of different tasks. * denotes the better performance (lower error) among the four tasks.

Vertical	MAE	MPE	RMSE	PCC_r
x_CHT_TM	1.291	0.911	2.175	0.995
y_CHT_TM	1.659	1.153	2.190	0.992
x_CHT_ACM	1.665	1.147 *	2.237 *	0.996
y_CHT_ACM	2.015	1.367	2.786	0.984
**Horizontal**	**MAE**	**MPE**	**RMSE**	**PCC_r**
x_CHT_TM	1.020 *	0.726 *	1.553 *	0.997
y_CHT_TM	1.672	1.184	2.520	0.964
x_CHT_ACM	2.128	1.516	3.014	0.988
y_CHT_ACM	2.298	1.588	3.489	0.934
**Circular**	**MAE**	**MPE**	**RMSE**	**PCC_r**
x_CHT_TM	1.755	1.260	2.925	0.985
y_CHT_TM	2.054	1.386	3.039	0.993
x_CHT_ACM	1.747	1.273	2.416	0.990
y_CHT_ACM	2.009	1.347 *	3.086	0.993
**Fixation**	**MAE**	**MPE**	**RMSE**	**PCC_r**
x_CHT_TM	1.537	1.353	2.182	0.999
y_CHT_TM	1.276 *	0.922 *	1.665 *	0.947
x_CHT_ACM	1.428 *	1.186	2.246	0.996
y_CHT_ACM	1.997 *	1.427	2.685 *	0.850

**Table 5 diagnostics-15-01446-t005:** The best performer/the worst performer among the four tasks for all the metrices. The best performer means the lowest error or the highest Pearson Correlation Coefficient value, and the worst performer is the opposite.

	MAE	MPE	RMSE	PCC_r
x_CHT_TM	Horizontal/Circle	Horizontal/Fixation	Horizontal/Circle	Fixation/Circle
y_CHT_TM	Fixation/Circle	Fixation/Circle	Fixation/Circle	Circle/Fixation
x_CHT_ACM	Fixation/Horizontal	Vertical/Horizontal	Vertical/Horizontal	Vertical/Horizontal
y_CHT_ACM	Fixation/Horizontal	Circle/Horizontal	Fixation/Horizontal	Circle/Fixation

**Table 6 diagnostics-15-01446-t006:** The performance of both algorithms on subjects with different iris colours. * denotes the better performance (lower error) among three subjects.

Subject 1	MAE	MPE	RMSE	PCC_r
x_CHT_TM	0.497 *	0.343 *	0.789 *	0.999
y_CHT_TM	1.325 *	0.945 *	1.974	0.992
x_CHT_ACM	1.182 *	0.807 *	1.746 *	0.996
y_CHT_ACM	1.210 *	0.847 *	1.832 *	0.993
**Subject 2**	**MAE**	**MPE**	**RMSE**	**PCC_r**
x_CHT_TM	2.791	1.985	3.732	0.972
y_CHT_TM	2.814	1.759	3.686	0.973
x_CHT_ACM	2.528	1.802	3.291	0.981
y_CHT_ACM	3.689	2.329	4.695	0.924
**Subject 3**	**MAE**	**MPE**	**RMSE**	**PCC_r**
x_CHT_TM	1.822	1.493	2.512	0.988
y_CHT_TM	1.415	1.143	1.957 *	0.991
x_CHT_ACM	2.148	1.712	2.823	0.990
y_CHT_ACM	2.037	1.583	2.734	0.980

## Data Availability

The datasets used and/or analysed during the current study are available from the corresponding author on reasonable request.

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
