# Peer review of "Validation of an Eye-Tracking Algorithm Based on Smartphone Videos: A Pilot Study"

_diagnostics, 2025, doi:10.3390/diagnostics15121446_

Round 1

Reviewer 1 Report

Comments and Suggestions for Authors

The article presents the development and validation of an eye-tracking algorithm for analyzing images using a smartphone camera.
The algorithm can be used in the diagnosis and follow-up of diseases such as autism, Alzheimer's, and Parkinson's.
I have a few suggestions about the article.
1. The term article should be used instead of project in the abstract section.
2. The algorithms used in the article are known and old algorithms. It should be detailed why these methods are preferred.
3. Limitations are given in Section 1.2. Deep learning architectures are widely used for eye tracking nowadays. It should be stated why deep learning is not used in eye tracking. Because deep learning performs better than the algorithms used in the article (Especially hardware limitations should be stated). Or, in Section 4.1, the literature should be cited, and the reason why deep learning is not used should be detailed. (The literature given in Section 1.4 on Deep Learning is not sufficient.
4. There are many more efficient eye tracking algorithms in the literature. It should be mentioned in Section 1.5 that the algorithm is lightweight. It should be stated that it is suitable for smartphones.
"The review above of existing literature indicates that the proposed algorithm introduces a novel approach to cross-modal, smartphone-based eye tracking, particularly designed for low-resource settings and edge computing." It would be appropriate to write this statement more emphatically.
5. pval and Pearsons r in Table 2 should be explained. Add the mathematical expressions of the used metrics as a table to the article.
6. The obtained results should be compared with the literature in the discussion section. What are the differences of the proposed model from the studies in the literature? What are its advantages? A comparison table is required. (Compare with criteria such as speed, frame count, hardware requirements, and performance parameters.)

Author Response

Comments 1: The term article should be used instead of project in the abstract section.
Response 1: Thank you for pointing this out. We agree with this comment. Therefore, we have replaced the term ‘project’ with ‘study’ or ‘article’ on page 1 line 10, page 5 229, page 6 line 235, 241, page 12 line 393, page 15 line 460.

Comments 2: The algorithms used in the article are known and old algorithms. It should be detailed why these methods are preferred.
Response 2: Agree. We have added section 1.4 Deep Learning Methods and Hardware Limitations to support this choice on page 4 line 148-183. Despite hardware considerations, another key reason deep learning models are not preferred in this context is the mismatch between public eye-tracking datasets and cognitive experiment settings, which we explained in more details on page 16 line 515-530.

Comments 3: Limitations are given in Section 1.2. Deep learning architectures are widely used for eye tracking nowadays. It should be stated why deep learning is not used in eye tracking. Because deep learning performs better than the algorithms used in the article (Especially hardware limitations should be stated). Or, in Section 4.1, the literature should be cited, and the reason why deep learning is not used should be detailed. (The literature given in Section 1.4 on Deep Learning is not sufficient.
Response 3: Thank you for pointing this out. We agree with this comment. We have added section 1.4 Deep Learning Methods and Hardware Limitations to supplement the literature about Deep Learning and hardware limitations on page 4 line 148-183.

Comments 4: There are many more efficient eye tracking algorithms in the literature. It should be mentioned in Section 1.5 that the algorithm is lightweight. It should be stated that it is suitable for smartphones. 
"The review above of existing literature indicates that the proposed algorithm introduces a novel approach to cross-modal, smartphone-based eye tracking, particularly designed for low-resource settings and edge computing." It would be appropriate to write this statement more emphatically.
Response 4: Agree. We have, accordingly, modified this statement on page 5 line 217-223. 

Comments 5: pval and Pearsons r in Table 2 should be explained. Add the mathematical expressions of the used metrics as a table to the article.
Response 5: Thank you for pointing this out. We agree with this comment. Therefore, we have added the mathematical formulations of the metrics on page 9 line 350-358

Comments 6: The obtained results should be compared with the literature in the discussion section. What are the differences of the proposed model from the studies in the literature? What are its advantages? A comparison table is required. (Compare with criteria such as speed, frame count, hardware requirements, and performance parameters.)
Response 6: Thank you for this comment. As mentioned, the current eye-tracking research is dominated by deep learning approaches, which we agree with. Unfortunately, most of the traditional algorithms proposed have not undergone formal validation and reported the metrics you mentioned. Despite our efforts in the literature search, we were unable to identify a traditional, non-deep learning method that reports comparable metrics within the time limit of the revision period. We have raised this issue on page 16 line 506-511. We recognise this gap and believe that advancing validated, lightweight traditional algorithms is promising.

Reviewer 2 Report

Comments and Suggestions for Authors

The paper is about efficient eye tracking problem through a smartphone camera. The authors introduce a CHT_TM method to solve the problem. The paper is interesting, but its presentation needs minor improvements.

(1) in abstract and main text, “this project …” is not proper for a research paper. It is better to use this paper or this study.

(2) on page 9, how to compute the total frame number, that is how to get 633. Please explain.

(3) on page 12 and page 13, table 3, several the finger results are marked with *, which demotes better performance. Please explain why they are marked *, because they are not the lowest. Table 4 has the same problem. Why the result with * symbol is neither the largest nor the lowest.

(4) on page 14, “no significant differences”, the conclusion significant is based on a hypothesis test. However, the paper does not use hypothesis test, so it is better not to use this word.

(5) since the title says a pilot study, it is better to release the dataset and the related code to readers to facilitate the research development.

Author Response

Comments 1: in abstract and main text, “this project …” is not proper for a research paper. It is better to use this paper or this study.
Response 1: Thank you for pointing this out. We agree with this comment. Therefore, we have replaced the term ‘project’ with ‘study’ or ‘article’ on page 1 line 10, page 5 229, page 6 line 235, 241, page 12 line 393, page 15 line 460.

Comments 2: on page 9, how to compute the total frame number, that is how to get 633. Please explain.
Response 2: Agree. We have explained why the manual annotation is taken every 10 frames on page 9 line 341-344. We also mentioned this in the limitations section on page 17 line 573-577.

Comments 3: on page 12 and page 13, table 3, several the finger results are marked with *, which demotes better performance. Please explain why they are marked *, because they are not the lowest. Table 4 has the same problem. Why the result with * symbol is neither the largest nor the lowest.
Response 3: Thank you for pointing this out. We are sorry for the confusion it caused. Therefore, we have added the explanation that the comparison is between different conditions/among different tasks/among different subjects on page 13 line 421, page 14 line 433, page 15 line 456.

Comments 4: on page 14, “no significant differences”, the conclusion significant is based on a hypothesis test. However, the paper does not use hypothesis test, so it is better not to use this word.
Response 4: Agree. We have replaced the word ‘significant’ with ‘notable’ on page 16 line 500.

Comments 5: since the title says a pilot study, it is better to release the dataset and the related code to readers to facilitate the research development.
Response 5: Agree. The dataset and the algorithm will be available upon reasonable request. 

Reviewer 3 Report

Comments and Suggestions for Authors

diagnostics-3611913

Recommendation: Minor revision

The manuscript is relevant to the Joural Diagnostics and the research is meaningful, thorough and is well-written.

The algorithms compared include Circular Hough Transform (CHT) for example. These algorithms are well-lnown and are existential to the public domain.. 

May the reader have access to the application of the more specific CHT algorithm used or its derivation thereof.in comparison with the counterpart algorithm discussed.

Author Response

Comments 1: The algorithms compared include Circular Hough Transform (CHT) for example. These algorithms are well-known and are existential to the public domain.. 
Response 1: Thank you for your comment. We agree that the algorithms we used are quite common. We have added section 1.4 Deep Learning Methods and Hardware Limitations to justify our choice based on the accessibility across the smartphone models on page 4 line 148-183. Despite hardware considerations, another key reason deep learning models are not preferred in this context is the mismatch between public eye-tracking datasets and cognitive experiment settings, which we explained in more details on page 16 line 515-530.

Comments 2: May the reader have access to the application of the more specific CHT algorithm used or its derivation thereof.in comparison with the counterpart algorithm discussed.
Response 2: Agree. The dataset and the algorithm will be available upon reasonable request.

Round 2

Reviewer 1 Report

Comments and Suggestions for Authors

I recommend that the article be accepted in its current form.